# Genome-Wide Identification of GRAS Transcription Factors and Their Functional Analysis in Salt Stress Response in Sugar Beet

**DOI:** 10.3390/ijms25137132

**Published:** 2024-06-28

**Authors:** Xiaolin Hao, Yongyong Gong, Sixue Chen, Chunquan Ma, Huizi Duanmu

**Affiliations:** 1Engineering Research Center of Agricultural Microbiology Technology, Ministry of Education, Heilongjiang University, Harbin 150080, China; hljuhxl1022@163.com (X.H.); gongyyong@163.com (Y.G.); chqm@hlju.edu.cn (C.M.); 2Heilongjiang Provincial Key Laboratory of Ecological Restoration and Resource Utilization for Cold Region, School of Life Sciences, Heilongjiang University, Harbin 150080, China; 3Department of Biology, University of Mississippi, Oxford, MS 38677, USA; schen8@olemiss.edu; 4Heilongjiang Provincial Key Laboratory of Plant Genetic Engineering and Biological Fermentation Engineering for Cold Region, School of Life Sciences, Heilongjiang University, Harbin 150080, China

**Keywords:** sugar beet, GRAS, transcription factors, salt stress, bioinformatics

## Abstract

GAI-RGA-and-SCR (GRAS) transcription factors can regulate many biological processes such as plant growth and development and stress defense, but there are few related studies in sugar beet. Salt stress can seriously affect the yield and quality of sugar beet (*Beta vulgaris*). Therefore, this study used bioinformatics methods to identify GRAS transcription factors in sugar beet and analyzed their structural characteristics, evolutionary relationships, regulatory networks and salt stress response patterns. A total of 28 *BvGRAS* genes were identified in the whole genome of sugar beet, and the sequence composition was relatively conservative. According to the topology of the phylogenetic tree, BvGRAS can be divided into nine subfamilies: LISCL, SHR, PAT1, SCR, SCL3, LAS, SCL4/7, HAM and DELLA. Synteny analysis showed that there were two pairs of fragment replication genes in the *BvGRAS* gene, indicating that gene replication was not the main source of BvGRAS family members. Regulatory network analysis showed that BvGRAS could participate in the regulation of protein interaction, material transport, redox balance, ion homeostasis, osmotic substance accumulation and plant morphological structure to affect the tolerance of sugar beet to salt stress. Under salt stress, *BvGRAS* and its target genes showed an up-regulated expression trend. Among them, *BvGRAS-15*, *BvGRAS-19*, *BvGRAS-20*, *BvGRAS-21*, *LOC104892636* and *LOC104893770* may be the key genes for sugar beet’s salt stress response. In this study, the structural characteristics and biological functions of BvGRAS transcription factors were analyzed, which provided data for the further study of the molecular mechanisms of salt stress and molecular breeding of sugar beet.

## 1. Introduction

The GRAS transcription factor family is named after the three family members GA [1], RGA [2] and SCR [3]. The GRAS protein is generally composed of 400–700 amino acids, and its C-terminal sequence is highly conserved. It is generally composed of five typical domains, namely LHRI, VHIID, LHRI, PFYRE and SAW motifs [4]. In contrast, the N-terminus of GRAS proteins contains different intrinsically disordered regions, so that these proteins can specifically recognize ligands by changing the N-terminus structure. Therefore, the GRAS family exhibits functional diversity [5], including participating in gibberellin [6], light signal [7], rape [8] and other signal transduction pathways, regulating axillary buds [9], meristem growth [10], meristem division [11] and maintaining apical dominance [12] in response to abiotic stresses such as salt, drought and shading [13] and participating in the formation of male gametes [14]. According to the difference in the N-terminal domain, GRAS family proteins are generally divided into nine subfamilies: SCL3, SHR, PAT1, LISCL, DELLA, SCR, LAS, SCL4/7 and HAM [15]. At present, the GRAS gene family has been identified and analyzed in more than 30 plants, such as *Arabidopsis thaliana* [16], rice (*Oryza sativa*) [17], soybean (*Glycine max*) [18] and ginseng (*Panax ginseng*) [19].

GRAS transcription factors play a variety of functions in plant growth and development, such as gibberellin (GA) signal transduction, stem tip meristem maintenance and biotic or abiotic stress response [20]. The *Arabidopsis* SCL3 protein belongs to the SCL3 subfamily, which can maintain the normal function of the gibberellin pathway by weakening the DELLA inhibition effect in the endoderm of roots, and it participates in the regulation of root cell elongation during the growth of *Arabidopsis thaliana* [21]. The rice SCR subfamily gene *OsGRAS32* plays an important regulatory role in gibberellin metabolism [22]. Overexpression of the *SmGRAS3* gene in *Salvia miltiorrhiza* resulted in the inhibition of the synthesis of salicylic acid and gibberellin and the growth of *Salvia miltiorrhiza* hairy roots [23]. In addition, a large number of studies have shown that GRAS transcription factors are involved in the responses of plants to biotic and abiotic stresses. After silencing the rice GRAS gene *OsCIGR2* by RNAi technology, the level of cell death was significantly higher than that of wild-type materials [24]. Overexpression of *BnLAS* in *Arabidopsis thaliana* can enhance the drought tolerance of transgenic materials by reducing the leaf water loss rate and stomatal opening and synthesizing more leaf epidermal wax [25]. The expression of the SCL4/7 subfamily gene *PeSCL7* in *Populus euphratica* is up-regulated by drought and salt stress. Overexpression of the *PeSCL7* gene can improve the drought and salt tolerance of transgenic *Arabidopsis thaliana* [26]. Under salt and mannitol treatment conditions, the expression level of the tomato (*Solanum lycopersicum*) GRAS family gene *SlGARS7* was significantly up-regulated, and the *SlGARS7* gene overexpression material was more drought-tolerant and salt-tolerant than the wild-type material [27]. Through the whole-genome identification of the GRAS family in sweet potato (*Ipomoea batatas*) and cotton (*Gossypium hirsutum*), it was also found that a large number of genes were up-regulated under salt, drought, heat and cold stress conditions [28,29]. Overexpression of the *PeSCL7* gene in *Populus euphratica* enhances the drought and salt tolerance of transgenic *Arabidopsis* plants [30]. Overexpression of the *Vitis amurensis VaPAT1* gene can significantly improve the drought tolerance, cold tolerance and salt tolerance of *Arabidopsis thaliana* [31].

As one of the important sugar crops in China, sugar beet has good cold resistance and a high sucrose content in its root tubers, and its sugar by-product molasses can produce betaine, methanol, ethanol, glycerol, acetone and other chemicals [32]. The sugar residue filter mud can be used as fertilizer and has the effect of neutralizing soil free acid [33]. In addition to sugar, some sugar beet varieties can be used for food and feed and have broad application prospects. Due to the aggravation of soil salinization in agricultural land, soil fertility and agricultural productivity are seriously threatened [34]. A study of sugar beet found that its sugar yield was significantly impaired under salt stress [35]. Therefore, the identification and functional verification of salt-tolerance-related genes in sugar beet are very important, which can provide key genetic resources for the molecular breeding of sugar beet.

The purpose of this study was to identify and analyze the GRAS transcription factor family in sugar beet. The genome-wide identification of the *BvGRAS* gene was carried out by bioinformatics methods, and its gene structure, sequence characteristics, chromosome localization, promoter functional elements, evolutionary relationship and salt stress response mode were analyzed. This provided a theoretical reference for the further analysis of the biological function of the *BvGRAS* gene, accurately predicted the performance of hybrid offspring from different parents in different ecological environments to provide reliable information support and also provided gene resources for the molecular breeding of salt tolerance in sugar beet.

## 2. Results

### 2.1. BvGRAS TF Family Members

After the genome-wide identification and domain screening, 28 sugar beet *GRAS* genes were obtained and sorted according to their gene ID, named from *BvGRAS-1* to *BvGRAS-28* (Table 1). Expasy was used to calculate the sequence length, isoelectric point, molecular weight and instability coefficient of the BvGRAS protein. The results of the physicochemical properties analysis showed that the sequence length of the BvGRAS protein was between 427 (BvGRAS-3)–840 (BvGRAS-24) amino acids, the isoelectric point was between 5.02 (BvGRAS-28)–6.81 (BvGRAS-5), the molecular weight was between 46.73 (BvGRAS-3)–91.89 (BvGRAS-24) kDa and the instability coefficient was greater than 40, indicating that the structure of the BvGRAS protein was more unstable.

### 2.2. Structure Analysis of BvGRAS TF Sequence

The exon–intron structure of the *BvGRAS* gene was visualized using the GSDS online tool. The analysis found that the *BvGRAS* gene basically did not contain an intron structure, and only *BvGRAS-24* contained an intron, and *BvGRAS-19* and *BvGRAS-14* had no upstream and downstream non-coding regions (Figure 1c). In addition, the MEME and SMART online databases were used to analyze the conserved motifs and conserved domains of the BvGRAS protein (Figure 1b). It was found that most of the BvGRAS proteins, except BvGRAS-13 and BvGRAS-27, contained only a single GRAS domain, while BvGRAS-13 and BvGRAS-27 also contained DELLA and SCOP domains, respectively. Conserved motif analysis showed that the distribution of 10 motifs in different BvGRAS proteins was relatively consistent. Combined with the phylogenetic tree of BvGRAS (Figure 1a and Appendix A), it was found that the sequences in the same evolutionary branch had the same or similar motif composition. All the sequences contained motifs 8, 6, 9, 1 and 5, and combined with domain information, it can be speculated that these five motifs were the core parts of the GRAS domain, while motifs 8, 6, 9, 1, 3, 7, 4, 2 and 5 corresponded to the complete GRAS domain.

### 2.3. Phylogenetic Analysis of BvGRAS TF

Phylogenetic trees were constructed using AtGRASs and BvGRASs sequences. According to the topological structure of the phylogenetic tree and the classification method of AtGRASs, the BvGRASs were divided into nine subfamilies: LISCL, SHR, PAT1, SCR, SCL3, LAS, SCL4/7, HAM and DELLA (Figure 2). BvGRAS-14, BvGRAS-15, BvGRAS-16, BvGRAS-17, BvGRAS-18, BvGRAS-19, BvGRAS-23 and BvGRAS-28 are members of the LISCL subfamily, BvGRAS-5, BvGRAS-10 and BvGRAS-11 are members of the SHR subfamily and BvGRAS-1, BvGRAS-4, BvGRAS-8, BvGRAS-20, BvGRAS-25 and BvGRAS-26 are members of the PAT1 subfamily. BvGRAS-3 and BvGRAS-23 are members of the SCR subfamily, BvGRAS-6 is a member of the SCL3 subfamily, BvGRAS-9 and BvGRAS-22 are members of the LAS subfamily, BvGRAS-12 is a member of the SCL4/7 subfamily, BvGRAS-2, BvGRAS-7 and BvGRAS-27 are members of the HAM subfamily and BvGRAS-13 and BvGRAS-21 are members of the DELLA subfamily.

### 2.4. BvGRAS TF Chromosome Localization and Collinearity Analysis

Chromosome localization analysis showed that the *BvGRAS* genes were unevenly distributed on each chromosome (Figure 3). There were seven *BvGRASs* genes on chromosome 7, five *BvGRASs* genes on chromosome 6, three *BvGRASs* genes on chromosomes 4 and 9 and two *BvGRASs* genes on chromosomes 1, 2, 5 and 8.

The MCScanX analysis method was used to obtain collinearity genes between species and explore whether the *GRAS* gene of the species experienced a gene replication event (Figure 3). The analysis found that the *BvGRASs* gene itself had only two pairs of collinearity genes (*BvGRAS-8~BvGRAS-20* and *BvGRAS-18~BvGRAS-23*). *AtGRASs* itself had 10 pairs of collinearity genes, and there were 19 pairs of collinearity genes between *AtGRASs* and *BvGRASs.* Therefore, *GRAS* genes generally showed more collinearity between species than the species themselves. In addition, the Ka/Ks analysis of these collinear genes that have undergone replication events (Appendix A) showed that the Ka/Ks values of all the gene pairs were far less than 1, indicating that these genes experienced strong purification selection during evolution.

### 2.5. BvGRAS TF Salt Stress Response Analysis and Co-Expression Analysis 

The transcriptome data of the sugar beet M14 strain under 200 mM and 400 mM salt stress were determined in the laboratory, and the transcriptome data of diploid sugar beet under 300 mM salt stress were downloaded from the SRA public database to analyze the expression pattern of *BvGRASs* under salt stress. The expression ratio of the salt treatment group and the control group was treated with LOG2, and the genes with LOG2 > 1 or <−1 were considered to be differentially expressed genes. The analysis showed that most of the *BvGRAS* genes were differentially expressed under salt stress (Figure 4). Among them, *BvGRAS-5*, *BvGRAS-7*, *BvGRAS-8*, *BvGRAS-9*, *BvGRAS-10*, *BvGRAS-19*, *BvGRAS-21* and *BvGRAS-26* were differentially expressed by more than four times, indicating that they can respond to salt stress.

In order to construct the regulatory network mediated by *BvGRAS*, the downstream regulatory genes of *GRAS* and the microRNAs that may be bound upstream were analyzed. The genes containing *GRAS* binding sites were screened in the whole genome of sugar beet, and their expression data in the transcriptome were obtained to calculate the correlation coefficient between *BvGRAS* and the expression of these genes (Appendix A). The results showed that most of the genes were strongly correlated with the expression of *BvGRAS*, and the genes with a correlation coefficient ≥ 0.8 were selected as the target genes of BvGRAS. The psRNATarget database was used to predict microRNAs that could bind to sugar beet GRAS transcription factors. The above data were used to construct a BvGRAS-mediated regulatory network, and the results are shown in Figure 5. In the regulatory network, genes such as *BvGRAS-14* and *BvGRAS-24* were core genes with a high connectivity and interacted with multiple target genes. *BvGRAS-8*, *BvGRAS-22*, *BvGRAS-24*, *BvGRAS-1* and *BvGRAS-27* were regulated by multiple microRNAs, indicating that these genes may be inhibited by microRNA cleavage or translation, thus exerting their biological functions. The analysis of the gene functions in the regulatory network showed that the target genes of BvGRAS transcription factors were involved in multiple gene families and were widely involved in various biological processes such as redox, protein interaction, transcriptional activation and translation activation. However, studies have shown that these genes are involved in the responses of plants to salt stress. Therefore, it is speculated that BvGRAS transcription factors and their upstream microRNAs participate in various ways to alleviate the effects of salt stress on the growth and development of sugar beet by regulating the expression of functional genes.

### 2.6. Promoter Analysis and qRT-PCR Analysis 

The 2000 bp upstream sequence of *BvGRASs* and its target gene CDS was extracted as a potential promoter sequence and submitted to the PlantCARE online database for promoter element analysis. The predicted elements were classified according to their functions, and a total of nine types of elements were obtained. The results are shown in Figure 6. The analysis showed that the promoter regions of *BvGRASs* were mostly anaerobic induction elements and functional elements responding to methyl jasmonate and abscisic acid, and the functional elements of their target genes also had similar trends. In addition, these genes also contained salicylic acid, auxin, gibberellin and defense stress response elements, indicating that these genes may be involved in a variety of abiotic stress response pathways and were mainly involved in methyl jasmonate and abscisic acid signal transduction pathways.

In order to explore the response mode of BvGRASs and their target genes to salt stress, sugar beet was treated with 200 mM salt stress, and root tissues were collected at 3 h, 6 h, 9 h and 12 h for qRT-PCR quantitative analysis. After removing genes that could not be designed specific primers, 21 *BvGRASs* genes and 5 target genes were quantitatively analyzed (Appendix A), and the results were as follows (Figure 7 and Figure 8). The results showed that except for *BvGRAS-17*, the other *BvGRAS* genes showed an up-regulated expression trend under salt stress, among which *BvGRAS-2*, *BvGRAS-19*, *BvGRAS-20* and other genes were specifically highly expressed. It is speculated that these genes are highly sensitive to salt stress and play an important biological function in the process of stress response. The analysis from the time point showed that most of the *BvGRASs* genes were up-regulated after 6 h of salt stress, and the expression level was down-regulated or did not change significantly at 3 h. Only *BvGRAS-1*, *BvGRAS-6*, *BvGRAS-14*, *BvGRAS-19*, *BvGRAS-21* and seven other genes were significantly up-regulated at 3 h. The five target genes had similar expression trends, with the peak expression at 6 h and a significant high expression at 12 h, indicating that these genes mainly responded to salt stress at these two time points.

### 2.7. Functional Annotation of BvGRAS TF

The GO functional annotation and enrichment analysis found that (Figure 9) most of the *BvGRAS* genes are enriched in biological processes, among which the highly enriched projects focus on endoderm cells, trichome patterns and leaf development regulation. It is worth noting that some genes are directly enriched in the cell response to salt, indicating that the *BvGRAS* gene may be involved in the regulation of the growth and development of sugar beet trichomes and leaves and directly involved in the response of sugar beet cells to salt stress. The KEGG annotation showed that *BvGRAS* may be involved in environmental information processing and signal transduction of plant hormones (Appendix A). Among them, *BvGRAS-1*, *BvGRAS-6*, *BvGRAS-8* and other genes (DELLA) are involved in the response pathway of gibberellin signaling, regulated by *GID1* and ubiquitinated by *GID2*, thereby regulating downstream transcription factors and functional genes and playing a role in plant stem growth and induced germination (Appendix A).

## 3. Discussion

With the improvement in genome sequencing technology, gene family identification has been carried out in more and more species. GRAS transcription factors have also been identified and functionally studied in various crops and cash crops, and the molecular mechanism of plant GRAS transcription factors in response to environmental stress has been discovered. However, there has been no report on related research in sugar beet. In this study, 28 GRAS transcription factors were identified from the sugar beet genome by bioinformatics methods, which were unevenly distributed on nine chromosomes. There was little difference in the molecular weight, pI and other indicators between the genes, and they also had a similar gene structure. Except for *BvGRAS-24*, the other members did not contain an intron structure. In addition, the conserved domain of the BvGRAS transcription factor was also consistent with the conserved motif composition. All sequences contained motifs 8, 6, 9, 1 and 5, so it is speculated that these five motifs are the core structure of the GRAS domain. In addition to the GRAS domain, BvGRAS-13 and BvGRAS-27 also contained the DELLA domain and SCOP domain, respectively, which may play more complex biological functions (Figure 1). The collinearity analysis showed that the intraspecific collinearity of the *GRAS* genes in sugar beet and *Arabidopsis* was weaker than that of the interspecific collinearity, indicating that *GRAS* genes in different species were not mainly derived from the gene replication of the species’ own genome (Figure 3) [36]. As orthologous genes, *GRAS* genes have a certain diversity in the common ancestors of these species and are distributed to different species as species differentiate [37].

A total of 66 *GRAS* genes were identified in ginger, and 57 *GRAS* genes were identified in rice. A total of 47 *GRAS* genes were identified in Dendrobium catenatum, and its *GRAS* gene has undergone polyploidization and its gene duplication events have evolved. In sugar beet, the *GRAS* genes did not undergo polyploidization, and the genes evolved from direct ancestral orthology [38]. In order to analyze the evolutionary relationship of GRAS transcription factors in different species, *Arabidopsis thaliana* AtGRAS and sugar beet BvGRAS were selected to construct a phylogenetic tree, and the BvGRAS subfamily was classified according to the research results in *Arabidopsis thaliana*. The results showed that the GRAS transcription factors could be divided into nine subfamilies, namely, LISCL, SHR, PAT1, SCR, LAS, SCL4/7, HAM, DELLA and SCL3 (Figure 2). In a study of *Arabidopsis thaliana*, the LISCL subfamily was found to be responsible for regulating or activating the transcription process related to plant stress response, regulating the formation of adventitious roots in response to auxin [39,40]. The SHR subfamily is related to root radial configuration, root growth, cell division and nodule development [41,42]. The PAT1 subfamily can regulate phyA-specific signal transduction, which is a positive regulator of phyB-dependent red light signal transduction and an early regulator of plant defense signal transduction [43]. The SCR subfamily is related to root growth and asymmetric cell division [44,45]. The LAS subfamily can regulate the formation of axillary buds and axillary meristems [46]. The SCL4/7 subfamily is a regulator of transcription processes related to environmental stress responses such as salt, osmotic and drought stress [47]. The HAM subfamily can maintain the bud meristem and is a transcriptional inhibitor of auxin response [48,49]. The DELLA subfamily is a repressor of GA response and can regulate environmental signals such as GA [50,51,52,53]. SCL3 can positively regulate the GA response pathway during root cell elongation and is an integration factor of GA/DELLA signaling and the SCR/SHR pathway [54].

Combined with the functional annotation results, it was found that the genes enriched in the salt stress response process were *BvGRAS-15*, *BvGRAS-16*, *BvGRAS-18*, *BvGRAS-20* and *BvGRAS-28* (Figure 9). These genes belong to the LISCL and PAT1 subfamilies, and the biological functions of these two subfamilies are to regulate plant stress response and defense signal transduction, which are closely related to salt stress response. Combined with previous gene function studies, it was found that the homologous genes *BvGRAS-15*, *BvGRAS-16*, *BvGRAS-18*, *BvGRAS-20* and *BvGRAS-28* could not only improve plant salt tolerance but also regulate cold stress by responding to GA signals, scavenge reactive oxygen species in light stress response and respond to dehydration stress by regulating stomatal aperture and density [18,55,56]. *BvGRAS-15*, *BvGRAS-16*, *BvGRAS-18*, *BvGRAS-20* and *BvGRAS-28* may also regulate plant stress response and defense signal transduction, so it is speculated that each member of *BvGRAS* also has the biological functions of the corresponding subfamilies. In addition, combined with the results of the collinearity analysis and phylogenetic tree, most of the genes with multiple collinearity relationships belonged to the LISCL subfamily members, such as *BvGRAS-18*, *BvGRAS-23* and *AtSCL9*, *AtSCL11*, *AtSCL14*, *AtSCL30*, *AtSCL33*, etc., indicating that LISCL is a more conservative evolutionary branch of the GRAS transcription factor family and is highly homologous in different species. It may play an important role in plant adaptation to environmental stress.

In this experiment, the expression level of *BvGRAS* after the 200 mM NaCl treatment was quantitatively analyzed by qRT-PCR to explore the response mode of *BvGRAS* to salt stress. The results (Figure 7) showed that the BvGRAS family members showed an up-regulated expression trend under salt stress, which was consistent with research on other species [57,58]. According to the different time points of treatment, 3 h and 6 h were divided into an early response, and 9 h and 12 h were divided into a late response. The analysis found that except for *BvGRAS-6* and *BvGRAS-17*, the other members had higher expression levels at 12 h, and the genes (*BvGRAS-15*, *BvGRAS-16*, *BvGRAS-18*, *BvGRAS-20*, *BvGRAS-28*) enriched in the salt stress response process in the functional annotation showed a peak expression level at 12 h. This indicated that BvGRAS family members may play a major role in the late response to salt stress. Among them, *BvGRAS-19* and *BvGRAS-21* had higher expression levels at each time point, indicating that they may be used as key genes for sugar beet response to salt stress and play important biological functions in early and late response processes.

In order to explore the potential interaction between BvGRAS family members under salt stress, the upstream and downstream binding microRNAs and target genes were predicted, and the regulatory network of BvGRAS was constructed based on the correlation matrix of the expression level (Figure 5). The analysis of the domain composition (Appendix A) and potential functions of the target genes in the network revealed that *LOC104892636* is a protein that controls the redox state of cells, maintains oxidative stress resistance and regulates signal transduction pathways through redox post-translational modification to reduce the damage of reactive oxygen species to plants [59]. *LOC104892371* is a member of the ATP-binding cassette (ABC) transporter superfamily that uses the energy released by ATP hydrolysis to transport numerous substrates on biofilms [60]. Previous studies have reported that ABC transporters can respond to various abiotic stresses [61], and studies in rice have also found a differential expression of ABC transporters under salt stress [62]. These results indicate that ABC transporters play a key role in plant salt tolerance. *LOC104893770* contains multiple HAT domains. A study of high chlorophyll fluorescence protein 107 (*HCF107*) in Arabidopsis showed that HAT repeats may be involved in protein–protein interactions [63]. *HCF107* exhibits sequence-specific RNA binding and RNA remodeling activity, which may lead to the activation of target gene cluster translation [64].

*LOC104896069* belongs to the mitochondrial F1F0ATP synthase F(A)6 subgroup. *MtATP6* was induced to express in the early stage of salt stress, indicating that it may be used as an early response gene and nuclear regulator to promote the formation of F1F0ATP synthase complex in mitochondria to enhance ATP production and maintain ion homeostasis under stress conditions [65]. *LOC104896272* belongs to the choline/ethanolamine kinase family, which can mediate the conversion of choline to betaine by regulating the biosynthesis of choline, thus affecting the osmotic pressure of plants in stress environments [66,67]. A study found that the gene *CcEthKin*, encoding choline/ethanolamine kinase in pigeon pea, was induced to express under salt stress, indicating that the gene may be involved in the biosynthesis pathway of betaine and may enhance the salt stress tolerance of plants by regulating the betaine content [68]. LOC104906401 belongs to the LONGIFOLIA (LNG) protein. Studies in *Arabidopsis thaliana* have found that the *LNG* gene plays an important role in the elongation of longitudinal polar cells by controlling the turgor pressure activated by *XTH17* and *XTH24* so as to regulate leaf morphology. Experiments have also shown that the leaf length of *LNG* gene mutants is reduced [69]. Therefore, it is speculated that the gene in sugar beet may play a regulatory role in leaf shrinkage during salt stress and leaf rehydration during recovery. *LOC104893189* belongs to the *TPX2* gene family. The cDNA of *TPX2* encodes a cell-wall-related peroxidase involved in the modification of the tomato cell wall structure. It has been found that the *TPX2* gene can improve the salt tolerance of tomato, and overexpression of the *TPX2* gene in tobacco can improve the germination rate under salt stress [69]. The analysis of the expression level of the target genes in this study also found that (Figure 8) *LOC104892636* and other genes had a consistent expression trend and were significantly highly expressed at 6 h and 12 h. Therefore, these genes may perform the above functions under salt stress and regulate the tolerance of sugar beet to salt stress by regulating protein interactions, material transport, redox balance, ion homeostasis, osmotic substance accumulation and plant morphology.

The purpose of this study was to identify the whole genome of GRAS and to analyze and predict the structure and function of the gene family from an overall perspective, without verifying the functions of all the genes one by one but with predicting the expression pattern under stress at the transcription level without in-depth research. Secondly, most of the data used in this study were obtained from public databases, and the existing beet database is not very complete, so the data could be more perfect. In this study, 28 GRAS transcription factors were identified from the sugar beet genome by bioinformatics methods, which were unevenly distributed on nine chromosomes. RT-PCR validation of the *GRAS* genes and target genes showed that multiple members of the GRAS gene family may play a critical role in salt stress. In the previous stage, our group conducted a functional verification of this gene family member *GAI* in sugar beet and found that its overexpression in *Arabidopsis* can enhance the salt tolerance of *Arabidopsis* plants. The experimental results will be published later. This provides a basis for the further exploration of the biological functions of *BvGRAS*. In the future, the application of these genes to the molecular breeding process of sugar beet will provide a theoretical basis for cultivating more salt-tolerant sugar beets and promoting more perfect, accurate and efficient sugar beet breeding, which is expected to bring new changes to the development of agriculture in the world.

## 4. Materials and Methods

### 4.1. Plant Material Treatment

Sugar beet seeds were germinated in vermiculite after thiram disinfection and transferred to a hydroponics system after one week of age. The ambient temperature was 28 °C, the light–dark ratio was 16 h/8 h and the optical density was 450 μmol m^−2^ s^−1^. After four weeks of culturing, the hydroponic sugar beet was treated with salt. The NaCl concentrations were set to 0 mM and 200 mM, and the treatment times were 0 h, 3 h, 6 h, 9 h and 12 h. After the treatment, the root tissue of the sugar beet was collected and stored in a refrigerator at −80 °C for subsequent RNA extraction.

### 4.2. Identification of GRAS TF Family

The total protein data of sugar beet and the hmm model (PF03514) of GRAS transcription factor were obtained in NCBI (https://www.ncbi.nlm.nih.gov/, accessed on 24 September 2022) and InterPro (https://www.ebi.ac.uk/interpro/entry/pfam/PF03514/ accessed on 24 September 2022) [70]. The preliminary identification was carried out by the HMMER3.0 software [71], and the identification results were screened by SMART (https://smart.embl.de/ accessed on 24 September 2022) [72] and other tools to obtain the GRAS transcription factor in sugar beet. The sequence length, isoelectric point, molecular weight and instability coefficient of the GRAS protein were calculated by Expasy (https://web.expasy.org/ accessed on 24 September 2022) [73].

### 4.3. Structural Analysis of GRAS TF Sequence

The exon–intron location information of the GRAS transcription factor in sugar beet was extracted from the sugar beet genome annotation file and visualized using the GSDS online tool (http://gsds.g-ao-lab.org/index.php accessed on 24 September 2022) [74]. The domain composition and location information of the sugar beet GRAS transcription factors were obtained on the SMART website, and their conserved motifs were analyzed using the MEME online database (https://meme-suite.org/ accessed on 24 September 2022) [75] and visualized using TBtools [76].

### 4.4. GRAS TF Phylogenetic Analysis

The Arabidopsis GRAS transcription factor protein sequence was downloaded from the TAIR website (https://www.arabidopsis.org/ accessed on 24 September 2022), and the phylogenetic tree [77] was constructed together with the sugar beet GRAS transcription factor and grouped according to the classification method of *Arabidopsis thaliana*. The maximum likelihood method was used to construct the phylogenetic tree, and the Bootstrap value was set to 1000. After the construction was completed, the iTOL online program (https://itol.embl.de/ accessed on 24 September 2022) [78] was used to further improve the construction results.

### 4.5. GRAS TF Chromosome Localization and Collinearity Analysis

*Arabidopsis thaliana* was selected for collinearity analysis with sugar beet. The genomic sequences and GFF files of sugar beet and Arabidopsis were obtained from the NCBI and TAIR databases, and the chromosome length and chromosome position of the GRAS transcription factors of the two species were extracted. The MCScanX(2022) software [79] was used to obtain the collinearity genes between the species so as to analyze the gene replication events of GRAS transcription factors between the species. All data were visualized by TBtools.

### 4.6. GRAS TF Salt Stress Response Analysis and Co-Expression Analysis

In the early stage of the laboratory experiment, the transcriptome data of sugar beet treated with 200 mM and 400 mM salt stress were determined, and the transcriptome data of sugar beet treated with 300 mM salt stress were obtained in the SRA database. Based on the above data, the response mode of sugar beet GRAS transcription factors to salt stress was analyzed. Gene expression profiles were drawn using TBtools(2023.2.10), GraphPad Prism and other software.

The PlantTFDB database (http://planttfdb.gao-lab.org/ accessed on 24 September 2022) [80] was used to predict the genes with GRAS transcription factor binding sites in the sugar beet genome, and the expression levels of these genes were obtained in the transcriptome data to calculate their correlation with the expression levels of GRAS transcription factors. The genes with a correlation coefficient ≥ 0.8 were selected as the target genes of GRAS transcription factors. The psRNATarget database (https://w-ww.zhaolab.org/psRNATarget/analysis accessed on 24 September 2022) [81] was used to predict microRNAs that could bind to sugar beet GRAS transcription factors. The data of sugar beet microRNAs were obtained from the study of Li et al. [82]. Based on the above data, the expression network of microRNA, transcription factors and target genes was constructed.

### 4.7. GRAS TF Promoter Analysis and qRT-PCR Analysis

A 2000 bp fragment upstream of the GRAS transcription factor CDS in sugar beet was extracted and used as the promoter region. The PlantCARE website (http://bioinformatics.psb.ug-ent.be/webtools/plantcare/html/ accessed on 24 September 2022) [83] was used to predict the type and number of functional elements on the promoter.

The total RNA of the root tissue of the sugar beet control group (0 mM) and salt treatment group (200 mM) was extracted by the TRIzol method [84]. According to the instructions of a reverse transcription kit from TAKARA Biotechnology, the extracted RNA was reverse transcribed to obtain the cDNA sequence. Using the CDS sequence of the GRAS transcription factor in sugar beet as a template, the specific primers required for qRT-PCR were designed using Primer Premier 5, and the SYBR Green I detection method was used. An ABI Prism 7500 PCR system was used for qRT-PCR, and the relative expression level of the gene was calculated by the 2^−∆∆Ct^ method [85].

### 4.8. GRAS TF Function Annotation

The basic data required for annotation were obtained in the GO database (http://geneontolo-gy.org/ accessed on 24 September 2022) [86]. The TBtools tool was used to perform GO annotation, enrichment and visualization of sugar beet GRAS transcription factors. The gene ontology was divided into biological processes, cellular components and molecular functions. KEGG annotation of sugar beet GRAS transcription factors was performed in the KofamKOALA (https://www.genome.jp/tools/kofam-koala/ accessed on 24 September 2022) database [87], and the TBtools software was used to complete the further enrichment and pathway analysis.

### 4.9. Data Analysis

The relevant real-time PCR data were calculated and integrated using Excel, and a bar chart was drawn using the GraphPad Prism 9 software. Significant differences in the results were analyzed using IBM SPSS Statistics 23. During the experiment, all data were subjected to three biological replicates of significance analysis (different letters represent significant differences between each set of data, *p* < 0.05).

## 5. Conclusions

In this study, the GRAS transcription factor family members of sugar beet were identified and analyzed by bioinformatics methods and tools, and they were divided into nine subfamilies according to the phylogenetic tree. The collinearity analysis showed that *BvGRAS* was not mainly generated by gene replication events. The promoter and regulatory network analysis showed that *BvGRAS* could participate in a variety of biological processes related to salt stress response. According to the expression patterns of *BvGRAS* and its target genes, potential salt-tolerant functional genes were screened, such as *BvGRAS-15*, *BvGRAS-19*, *BvGRAS-20*, *BvGRAS-21*, *LOC104892636*, *LOC104893770*, etc. In addition, the microRNA analysis of the upstream of BvGRAS found that some microRNAs (mtr-miR171d, gma-miR171k-3p, PC-5p-160078_33, etc.) could bind to members such as *BvGRAS-1* and *BvGRAS-24*, thereby indirectly regulating functional genes such as *LOC104893770* and *LOC104896069*, indicating that the above microRNAs may have biological functions in regulating the salt stress response in sugar beet.

## Figures and Tables

**Figure 1 ijms-25-07132-f001:**
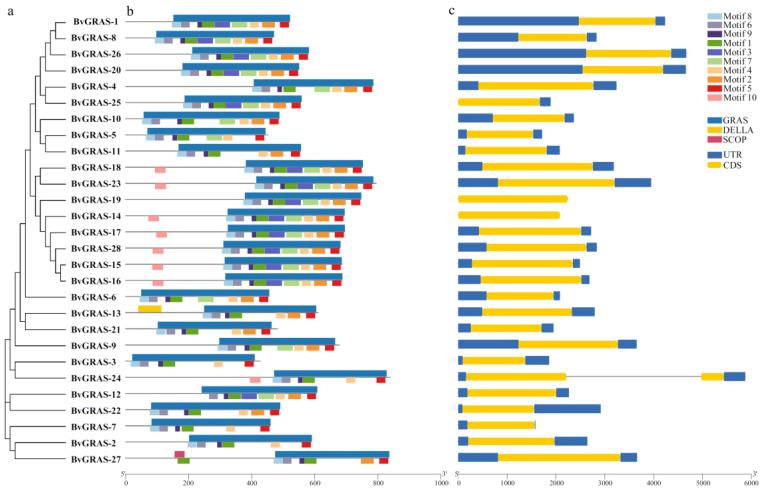
BvGRAS TF sequence structure. (**a**) BvGRAS TF phylogenetic tree. (**b**) Conserved motifs and conserved domains of BvGRAS TF; the upper half of each sequence is divided into conserved domains, and the lower half is divided into conserved motifs. (**c**) *BvGRAS* gene structure. Blue is the non-coding region, yellow is the exon and the black horizontal line indicates the intron.

**Figure 2 ijms-25-07132-f002:**
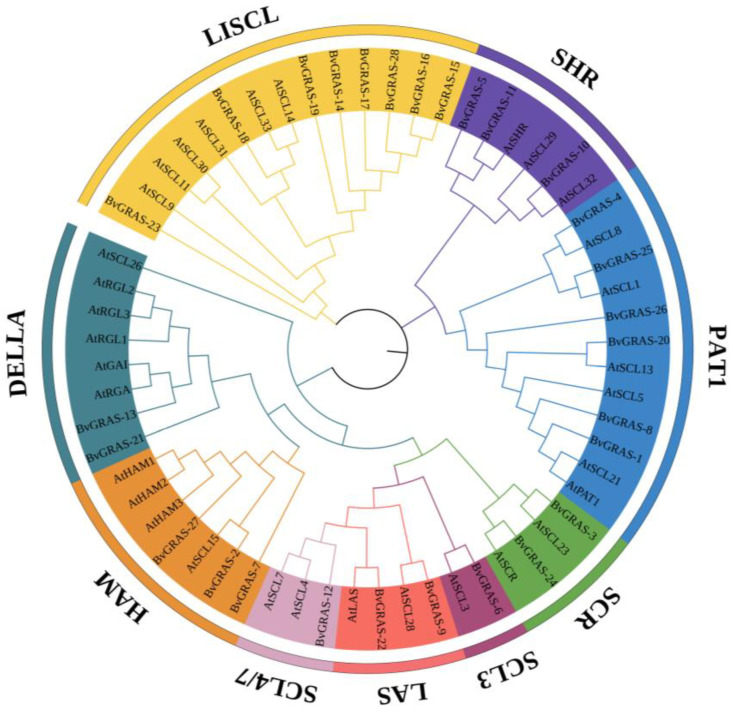
GRAS TF phylogenetic analysis. The phylogenetic tree based on the maximum likelihood method was constructed using AtGRASs and BvGRASs sequences. According to the classification method of *Arabidopsis thaliana*, BvGRASs can be divided into nine subfamilies, namely LISCL, SHR, PAT1, SCR, SCL3, LAS, SCL4/7, HAM and DELLA.

**Figure 3 ijms-25-07132-f003:**
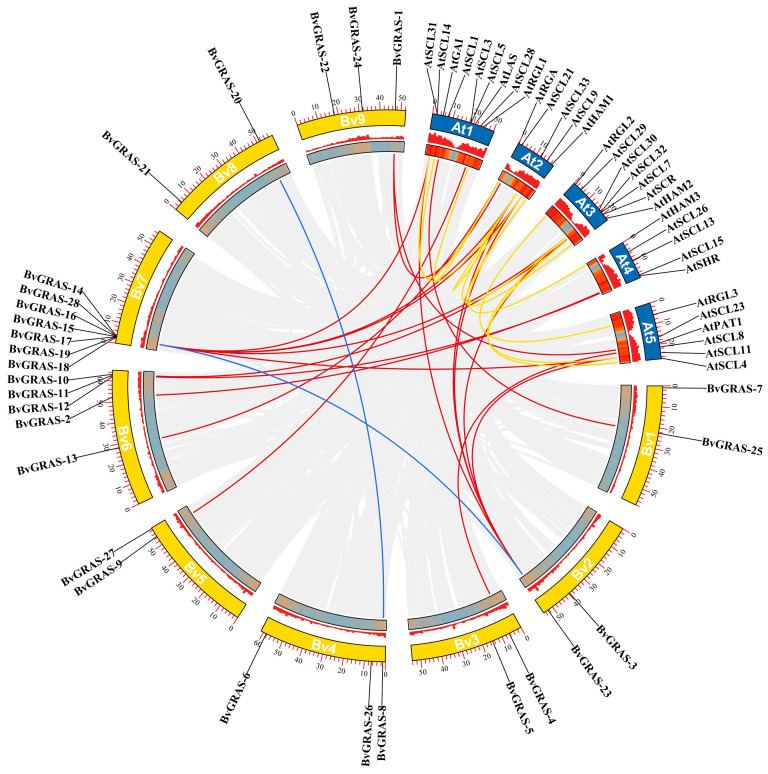
GRAS TF chromosome localization and collinearity analysis. The yellow box represents the sugar beet chromosome, and the blue box represents the *Arabidopsis* chromosome. The blue line represents the gene pair experiencing gene replication events between *BvGRAS* and *BvGRAS*, the yellow line represents the gene pair experiencing gene replication events between *AtGRAS* and *AtGRAS* and the red line represents the gene pair experiencing gene replication events between *AtGRAS* and *BvGRAS*.

**Figure 4 ijms-25-07132-f004:**
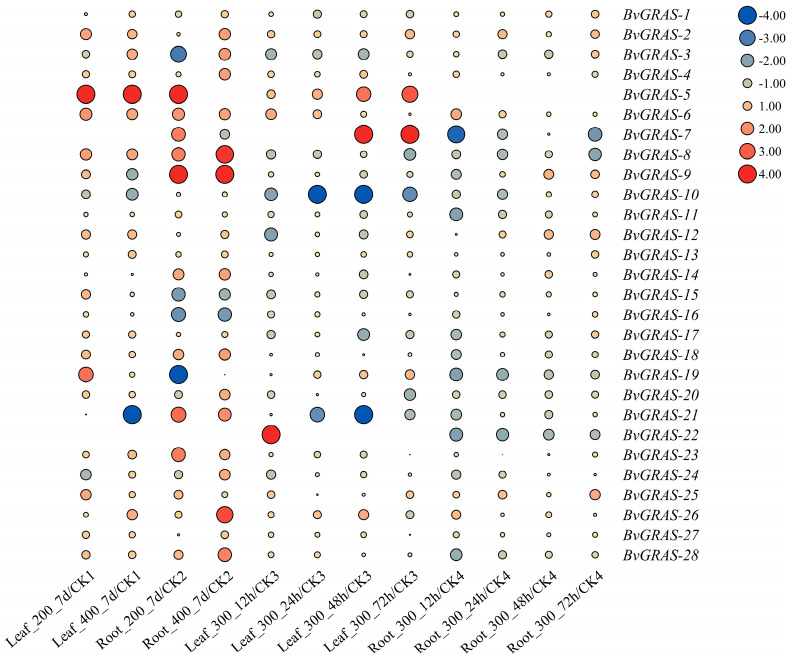
*BvGRAS* salt stress response pattern analysis. In the figure, red indicates up-regulated expression, blue indicates down-regulated expression and node size indicates differential expression multiples. Numbers 200, 400 and 300 indicate that the salt concentration was 200 mM, 400 mM and 300 mM. The time of salt treatment was expressed as 7 d, 12 h, 24 h, 48 h and 72 h. CK1: leaf tissue control group (0 mM), CK2: root tissue control group (0 mM), CK3: leaf tissue control group (0 h), CK4: root tissue control group (0 h).

**Figure 5 ijms-25-07132-f005:**
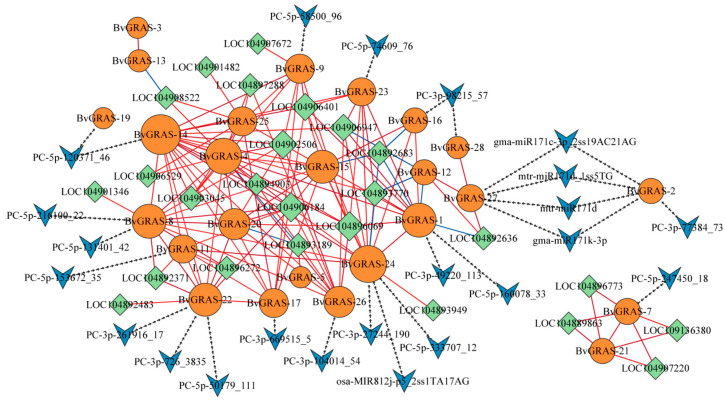
*BvGRAS* regulatory network. The orange circle represents *BvGRASs*, the green diamond represents *BvGRAS* downstream target genes and the blue arrow shape represents *BvGRAS* upstream microRNA. The arrow line indicates the regulatory relationship between microRNA and *BvGRAS*, the red line indicates that the gene expression was positively correlated and the blue line indicates that the gene expression was negatively correlated. The node size in the graph represents the size of the gene connectivity. The microRNA named PC was newly identified in sugar beet, and ss indicates that the microRNA sequence of sugar beet had mutation sites compared with other species. Taking 2ss19AC21AG as an example, there were two mutation sites, where A mutation at position 19 was C, and A mutation at position 21 was G.

**Figure 6 ijms-25-07132-f006:**
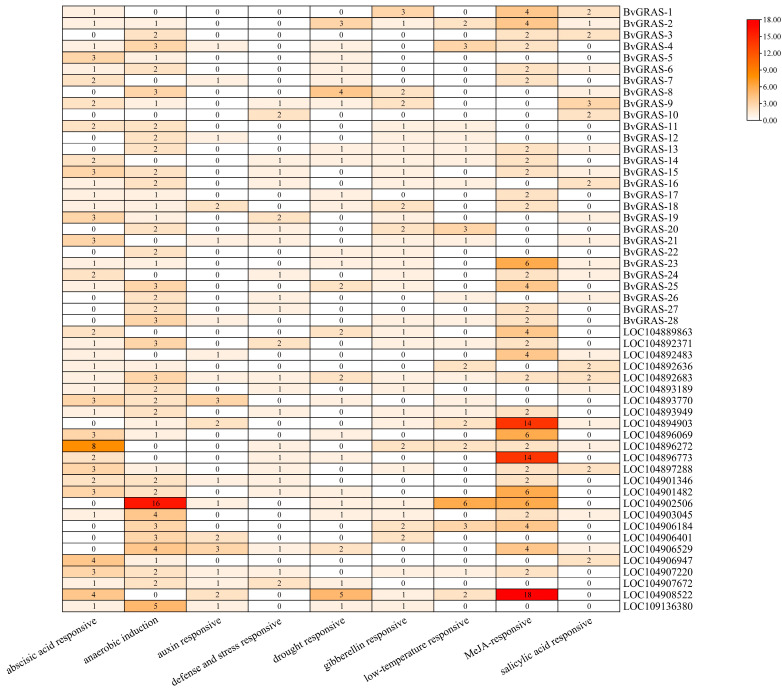
Prediction results of functional elements in BvGRASs and its target gene promoter region. A total of nine functional elements were predicted, including abscisic acid response, anaerobic induction, auxin response, defense stress response, drought response, gibberellin response, low temperature response, methyl jasmonate response and salicylic acid response. The color in the figure represents the number of functional elements contained in the gene.

**Figure 7 ijms-25-07132-f007:**
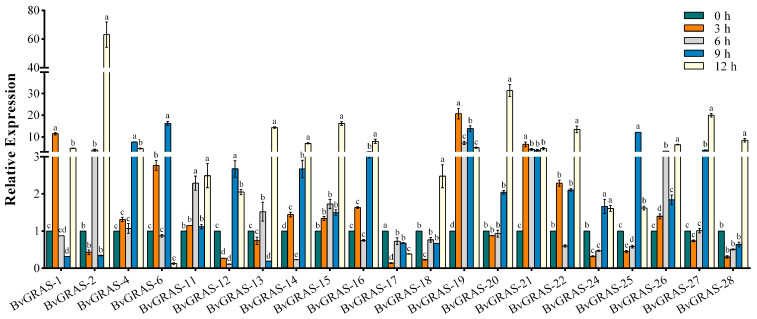
Expression pattern of *BvGRAS* under salt stress. The root tissues of sugar beet treated with 200 mM NaCl were analyzed by qRT-PCR. The sampling times were 0, 3, 6, 9 and 12 h, respectively. Each treatment corresponded to three biological and technical repetitions, and the control group was not treated with NaCl. The relative expression levels of 21 *BvGRAS* genes were calculated by 2^−∆∆^Ct method. The expression level at 0 h was used as a reference (relative expression level was 1). A result of less than 1 was down-regulated, and a result of more than 1 was up-regulated. The data were analyzed by Duncan’s analysis of variance, and different lowercase letters indicate differences in expression.

**Figure 8 ijms-25-07132-f008:**
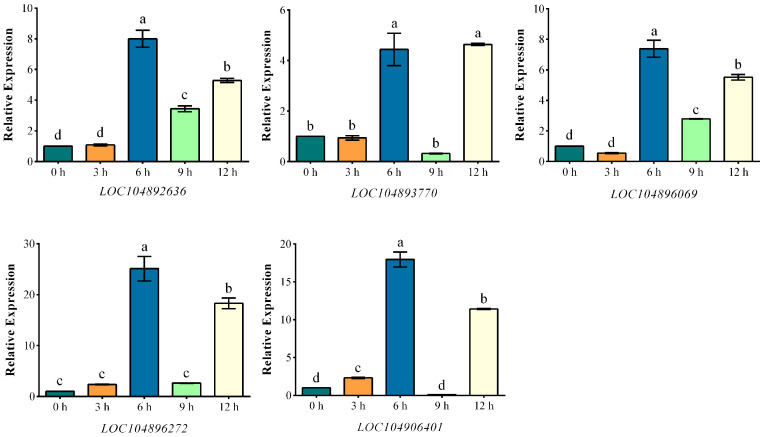
The expression pattern of *BvGRAS* transcription factor target genes under salt stress. The root tissues of sugar beet treated with 200 mM NaCl were analyzed by qRT-PCR. The sampling times were 0, 3, 6, 9 and 12 h, respectively. Each treatment corresponded to three biological and technical repetitions, and the control group was not treated with NaCl. The relative expression levels of target genes were calculated by 2^−∆∆^Ct method. The expression level at 0 h was used as a reference (the relative expression level was 1). A result of less than 1 was down-regulated expression, and a result of more than 1 was up-regulated expression. The data were analyzed by Duncan’s analysis of variance, and different lowercase letters indicate differences in expression.

**Figure 9 ijms-25-07132-f009:**
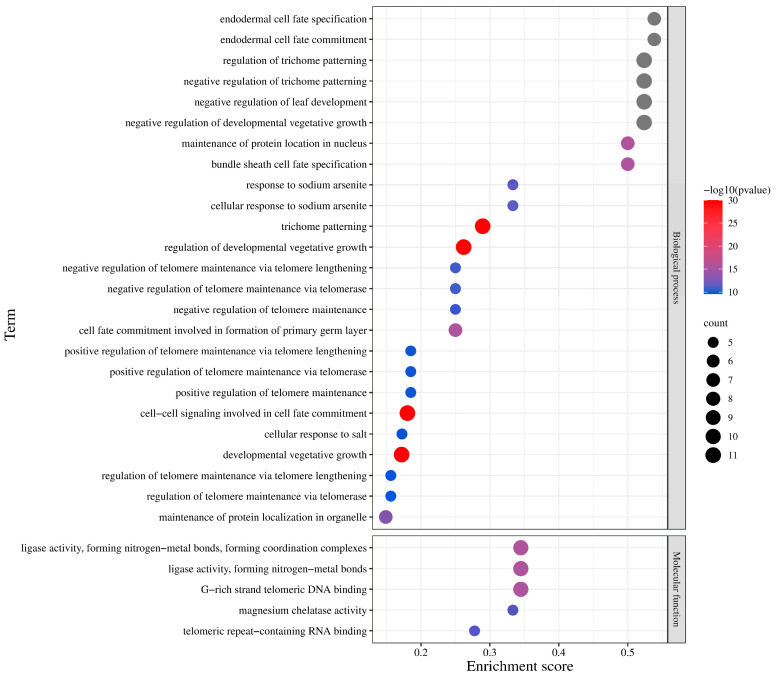
GO enrichment analysis of *BvGRAS* gene. The horizontal axis represents the enrichment degree of genes in the corresponding projects, and the vertical axis is the 30 projects with the highest enrichment degree of *BvGRAS* genes. The size of the circle represents the number of genes enriched in the corresponding project. The color from blue to red indicates the *p*-value from large to small, and the color gray indicates that the *p* value was 0.

**Table 1 ijms-25-07132-t001:** Basic information of BvGRAS family.

Gene ID	Gene Name	Chromosome	Location	Length (aa)	pI	MW (kDa)	Instability Index
LOC104883303	BvGRAS-1	Chr9	47272109–47276339	521	5.84	58.36	46.25
LOC104884709	BvGRAS-2	Chr6	52483906–52486547	589	6.22	64.61	44
LOC104886598	BvGRAS-3	Chr2	39617378–39619237	427	6.11	46.73	46.8
LOC104888390	BvGRAS-4	Chr3	6251492–6254730	783	5.58	86.54	52.97
LOC104888856	BvGRAS-5	Chr3	14855492–14857205	452	6.81	51.96	44.14
LOC104890059	BvGRAS-6	Chr4	58393359–58395438	456	6.04	50.93	54.7
LOC104891471	BvGRAS-7	Chr1	943670–945249	460	6.21	52.16	40.82
LOC104891820	BvGRAS-8	Chr4	1276279–1279109	466	5.29	52.07	41.11
LOC104892338	BvGRAS-9	Chr5	54516753–54520401	678	5.67	76.68	62.07
LOC104894934	BvGRAS-10	Chr6	63135438–63137801	488	5.59	54.88	42.41
LOC104894957	BvGRAS-11	Chr6	62691123–62693201	555	6.12	63.65	48.38
LOC104895016	BvGRAS-12	Chr6	61969338–61971599	605	5.08	66.73	48.26
LOC104896788	BvGRAS-13	Chr6	28255757–28258546	611	5.37	67.05	44.34
LOC104900088	BvGRAS-14	Chr7	3155937–3158021	694	5.69	78.84	52.41
LOC104900090	BvGRAS-15	Chr7	3130525–3133015	687	5.32	77.41	47.5
LOC104900091	BvGRAS-16	Chr7	3135597–3138281	687	5.22	77.37	47.8
LOC104900093	BvGRAS-17	Chr7	3118322–3121040	696	5.46	78.81	43.7
LOC104900094	BvGRAS-18	Chr7	3090537–3093719	752	5.49	84.21	49.33
LOC104900447	BvGRAS-19	Chr7	3104795–3107038	747	5.29	84.39	52.57
LOC104900825	BvGRAS-20	Chr8	51348137–51352794	548	5.84	61.54	52.03
LOC104902119	BvGRAS-21	Chr8	4394976–4396924	482	5.22	53.32	46.77
LOC104903785	BvGRAS-22	Chr9	18250458–18253370	488	6.23	55.33	61.09
LOC104904278	BvGRAS-23	Chr2	52598520–52602465	795	6.08	89.47	49.86
LOC104904740	BvGRAS-24	Chr9	31592859–31598730	840	5.82	91.89	56.06
LOC104905710	BvGRAS-25	Chr1	22511536–22513426	556	5.1	62.85	46.48
LOC104907103	BvGRAS-26	Chr4	6676833–6681498	579	6.17	64.80	59.03
LOC104907825	BvGRAS-27	Chr5	58827250–58830908	835	5.85	91.33	51.8
LOC109133554	BvGRAS-28	Chr7	3146288–3149122	681	5.02	76.76	46.62

## Data Availability

All the raw data as well as gene annotations can be found in the National Center for Biotechnology Information (https://www.ncbi.nlm.nih.gov/, accessed on 24 September 2022). All other data are available from the corresponding author upon reasonable request.

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
