# Peer review of "Genome-Wide Identification of GRAS Transcription Factors and Their Functional Analysis in Salt Stress Response in Sugar Beet"

_ijms, 2024, doi:10.3390/ijms25137132_

Round 1

Reviewer 1 Report

Comments and Suggestions for Authors

In the present manuscript, the authors have investigated the Genome-wide identification of GRAS transcription factors and their functional analysis in salt stress response in sugar beet. Nonetheless, there is still a significant level of ambiguity in the description and figure. I'll list some of my primary concerns below.

1)    Protein named should not be provided in italic format. But gene names must be in italic.

2)    In the section that talks about quantitative real-time PCR (qRT-PCR) analysis, it'd help if the authors could throw in more details about the stats tests used to sort through expression data. Like, did ya use two-tailed tests and what was the significance level you were lookin' at (e.g., p<0.05)? This could really make your results clearer and more trustworthy.

3)    Functional validation of GRAS genes is required.

4)    What are the potential applications of genomic insights into GRAS genes for sugar beet breeding programs?

5)    In the section “Identification of GRAS TF family” authors are requested to provide Pfam ID.

6)    In Figure 1. authors are requested to provide motif sequences.

7)    a comparative analysis of GRAS genes with those from other plant species could provide valuable evolutionary insights.

Author Response

Point 1: Protein named should not be provided in italic format. But gene names must be in italic.

Response 1: Thank you for your reminder. I have midofied it in line 243, 260, 286, 289, 291, 293, 294, 295 and 412.

Point 2: In the section that talks about quantitative real-time PCR (qRT-PCR) analysis, it'd help if the authors could throw in more details about the stats tests used to sort through expression data. Like, did ya use two-tailed tests and what was the significance level you were lookin' at (e.g., p<0.05)? This could really make your results clearer and more trustworthy.

Response 2: Thank you for your suggestion. I have added it in line 526-531.

Point 3: Functional validation of GRAS genes is required.

Response 3: Thank you for your suggestion. The purpose of this study was to identify the whole genome of GRAS gene family, analyze its structure and predict its function from an overall perspective, so the function of all genes was not verified one by one. However, our research group has conducted functional verification of GAI in GRAS gene family in the early stage, and found that this gene can improve the tolerance of sugar beet to salt stress. The results of this experiment will be published later. This section is mentioned in lines 437-445 of the text.

Point 4: What are the potential applications of genomic insights into GRAS genes for sugar beet breeding programs?

Response 4: Thank you for your suggestion. I have added it in line 441-445.

Point 5: In the section “Identification of GRAS TF family” authors are requested to provide Pfam ID.

Response 5: Thank you for your suggestion. I have added it in line 457-459.

Point 6: In Figure 1. authors are requested to provide motif sequences.

Response 6: Thank you for your suggestion. I have added it in figure S1.

Point 7: a comparative analysis of GRAS genes with those from other plant species could provide valuable evolutionary insights.

Response 7: Thank you for your reminder. I have added it in line 325-329.

Reviewer 2 Report

Comments and Suggestions for Authors

In this paper, researchers identify GRAS TF under salt stress in beet. Overall, the introduction is sufficient, the results are well presented and, the discussion corresponds to the results without much speculation. Few mistakes or references to be added, check the attached file.

Question: did the authors confirm any transcriptomic analysis by qPCR sample of plants that underwent stress and others that didn't?

Author Response

Point 1: In this paper, researchers identify GRAS TF under salt stress in beet. Overall, the introduction is sufficient, the results are well presented and, the discussion corresponds to the results without much speculation. Few mistakes or references to be added, check the attached file.

Question: did the authors confirm any transcriptomic analysis by qPCR sample of plants that underwent stress and others that didn't?

Response 1: Thank you for your reminder. In Figure 7, beets after 0,3,6,9 and 12 h of NaCl treatment were sampled and analyzed for transcript level, where 0 h sampled samples without stress treatment.

Reviewer 3 Report

Comments and Suggestions for Authors

Dear authors,

I appreciate the work presented in this manuscript. Overall, it is well-organized and well-structured. The methodologies are elaborated with sufficient details, and the results section is also well-organized. However, I believe the discussion section requires some amendments. Please find my comments below. Overall, the manuscript is very nice (good job!!) and can be accepted.

1.    General comment: please provide the full names of all the proteins where they are first mentioned.

2.    Lines 48-50: The authors might consider mentioning some specific functions of the GRAS transcription factors such as gibberellin signaling, shoot apical meristem maintenance, stress resistance, etc.

3.    Lines 90-92: Very general implications. Please think more innovatively, about how the current study can be used in the future.

4.    Line 105: Please make it a main table, not supplementary.

5.    Please briefly discuss the following: the implications of the current study, its drawbacks, how it can advance the field of sugar beet research, and to what extent.

Author Response

Point 1: General comment: please provide the full names of all the proteins where they are first mentioned.

Response 1: Thank you for your reminder. I have midofied it in line 16.

Point 2: Lines 48-50: The authors might consider mentioning some specific functions of the GRAS transcription factors such as gibberellin signaling, shoot apical meristem maintenance, stress resistance, etc.

Response 2: Thank you for your suggestion. The specific functions of the GRAS transcription factors have been mentioned in lines 53-81.

Point 3: Lines 90-92: Very general implications. Please think more innovatively, about how the current study can be used in the future.

Response 3: Thank you for your suggestion. I have added it in line 428-445.

Point 4: Line 105: Please make it a main table, not supplementary.

Response 4: Thank you for your suggestion. I have turned line 105 into the main table.

Point 5: Please briefly discuss the following: the implications of the current study, its drawbacks, how it can advance the field of sugar beet research, and to what extent.

Response 5: Thank you for your suggestion. I have added it in line 428-445.

Round 2

Reviewer 1 Report

Comments and Suggestions for Authors

I have no further suggestions and believe that the manuscript is now suitable for publication.